# Structural Insight into TNIK Inhibition

**DOI:** 10.3390/ijms232113010

**Published:** 2022-10-27

**Authors:** Mutsuko Kukimoto-Niino, Mikako Shirouzu, Tesshi Yamada

**Affiliations:** 1Laboratory for Protein Functional and Structural Biology, RIKEN Center for Biosystems Dynamics Research, 1-7-22 Suehiro-cho, Tsurumi-ku, Yokohama 230-0045, Japan; 2Department of Gastrointestinal and Pediatric Surgery, Tokyo Medical University, 6-7-1 Nishishinjuku, Shinjuku-ku, Tokyo 160-0023, Japan

**Keywords:** TNIK, inhibitor, colorectal cancer, Wnt signaling, APC, β-catenin, TCF4

## Abstract

TRAF2- and NCK-interacting kinase (TNIK) has emerged as a promising therapeutic target for colorectal cancer because of its essential role in regulating the Wnt/β-catenin signaling pathway. Colorectal cancers contain many mutations in the Wnt/β-catenin signaling pathway genes upstream of TNIK, such as the adenomatous polyposis coli (*APC*) tumor suppressor gene. TNIK is a regulatory component of the transcriptional complex composed of β-catenin and T-cell factor 4 (TCF4). Inhibition of TNIK is expected to block the aberrant Wnt/β-catenin signaling caused by colorectal cancer mutations. Here we present structural insights into TNIK inhibitors targeting the ATP-binding site. We will discuss the effects of the binding of different chemical scaffolds of nanomolar inhibitors on the structure and function of TNIK.

## 1. TNIK as a Druggable Target of Wnt Signaling

TNIK is a serine/threonine kinase of the germinal center kinase (GCK) family. The TNIK cDNA was initially cloned from the human brain and reported as the first GCK member potentially involved in cytoskeletal regulation [1]. Human TNIK is a 1360-residue protein consisting of an N-terminal kinase domain, an intermediate domain, and a C-terminal domain conserved in the GCK family, which is termed the citron homology (CNH) domain (Figure 1a). The intermediate domain mediates interactions with the adaptor proteins TRAF2 and NCK. Like other GCK kinases, the CNH domain of TNIK activates the c-Jun N-terminal kinase (JNK) pathway. The CNH domain of TNIK specifically interacts with Rap2, a small GTPase of the Ras family, and is involved in the regulation of the actin cytoskeleton [2].

Two subsequent studies identified TNIK as a protein that interacts with TCF4 in colorectal cancer cells [3,4]. TCF4 is a member of the TCF/LEF family of transcription factors and is activated only when bound to β-catenin [5]. Constitutive transcription of TCF4 target genes caused by aberrant Wnt signaling, such as loss of APC function, is critical for tumorigenesis [6].

TNIK is an essential activator in Wnt signaling [4,7], and directly binds to TCF4 and β-catenin via its kinase and intermediate domains, respectively (Figure 1a). TNIK specifically phosphorylates the conserved serine 154 residue of TCF4. Colorectal cancer growth is highly dependent on the kinase activity of TNIK. Thus, inhibition of the TNIK kinase activity by small molecule inhibitors is a promising strategy in colorectal cancer therapy.

TNIK, like other protein kinases, has a kinase domain structure with an ATP-binding site that historically has been targeted by small molecule inhibitors [8]. Therapeutic agents targeting the Wnt signaling pathway, including TNIK, were recently reviewed [9,10,11]. In this review, we will focus on the structural basis of novel small-molecule TNIK inhibitors. We will also briefly describe the conformational changes in the TNIK kinase domain upon inhibitor binding, in context with the full-length structure.

## 2. Overall Structure of the Kinase Domain

Structural investigations of TNIK have primarily focused on the kinase domain. The TNIK kinase domain has a two-lobed structure, consisting of N- and C-lobes, common to protein kinases (Figure 1b). The short segment connecting the two lobes is called the hinge region, and the deep cleft between the two lobes serves as an ATP-binding site. The assembly between these two lobes is flexible: a comparison of the protein kinase structures with and without ATP demonstrated that the N-lobe in the apo-form shifts away from the C-lobe, resulting in an open, inactive conformation [13]. Consistently, the crystal structure of the apo-form TNIK kinase domain revealed an inactive conformation with an open ATP-binding cleft [12]. Ligand binding tends to close the ATP-binding cleft, and thus stabilizes the active conformation of protein kinases.

## 3. Effects of Inhibitor Binding on the Structure of TNIK

Explorations of TNIK inhibitors began more than a decade ago, and increasing numbers of small molecule inhibitors with a variety of chemical skeletons have been discovered (Table 1 and Figure 2). All TNIK inhibitors reported so far target the ATP-binding site, and structure-based optimizations have been used to increase their potencies. Several TNIK structures bound to different inhibitors are available in the Protein Data Bank. Structural comparisons revealed that the inhibitors bind to two distinct states of the TNIK kinase domain: a closed, active conformation and an open, inactive conformation. Overall, the two states can be distinguished by the “in” and “out” conformations of the αC-helix in the N-lobe of the TNIK kinase domain, corresponding to the closed and open conformations, respectively (Table 1 and Figure 3). Below, we will describe the characteristics of the inhibitor-bound structures of the TNIK kinase domain in more detail.

### 3.1. Wee1Chk1 Inhibitor PD407824

The first structure of the TNIK kinase domain was determined using the phenylpyrrolocarbazole-based inhibitor PD407824 (PDB code: 2X7F, to be published). PD407824 was developed as an inhibitor of the checkpoint kinases Wee1 and Chk1 [29,30,31]. PD407824 was also shown to stably bind to GAK (cyclin G-associated kinase) and has been used in the crystallographic analysis of this kinase [32].

PD407824 binds to TNIK with an IC_50_ value of 0.7 nM [9]. The PD407824-bound TNIK structure revealed that the pyrrolocarbazole ring of PD407824 interacts with the hinge region of TNIK by forming multiple hydrogen bonds with the backbone carbonyl and amide groups of Glu106 and Cys108 (Figure 4a). The inhibitor also hydrogen bonds to Tyr36 (N-lobe) of TNIK. These interactions appear to stabilize PD407824 binding to TNIK, resulting in a closed, active conformation of the TNIK kinase domain (Figure 3a).

### 3.2. Phenylaminopyridine-Based Inhibitors

A series of 4-phenyl-2-phenylaminopyridine analogs have been identified as potent and selective inhibitors of TNIK [17]. One of the analogs, compound **3**, inhibited TNIK activity with an IC_50_ value of 6 nM and was highly selective for TNIK and MAK4K4 (90% identical kinase domains). However, unlike previous studies [4,7], these phenylaminopyridine-based TNIK inhibitors were less effective in inhibiting Wnt-driven gene expression. In addition, the inhibitors had no specific effect on the viability of Wnt-activated colorectal cancer cells. These results suggested that the kinase activity of TNIK is not essential for Wnt-activated colorectal cancer cells. The TCF-4 binding function of the TNIK kinase domain seems to be important, and probably could not be inhibited by phenylaminopyridine-based TNIK inhibitors.

Subsequently, the crystal structure of TNIK bound to one of the phenylaminopyridine analogs, compound **9** (PDB ID: 5AX9), was determined by a comparative structural analysis with NCB-0846 [12] (see next subsection). Examination of the compound **9**-binding site revealed that the phenylaminopyridine scaffold interacts with the hinge region of TNIK through two hydrogen bonds with the Cys108 backbone (Figure 4b). The cyano group of the 4-phenyl moiety extends deeply within the ATP-binding cleft and contacts Lys54, Met105, and Asp 171. Lys54 forms a salt bridge with Glu69, which is characteristic of the closed, active conformation of protein kinases.

### 3.3. NCB-0846

NCB-0846, a quinazoline analog, was identified as the first orally available small molecule TNIK inhibitor with anti-Wnt signaling activity [12]. NCB-0846 showed strong inhibitory activity against TNIK (IC_50_ = 21 nM). It inhibited colorectal cancer cell proliferation, colorectal cancer stemness (CSC), and tumorigenesis. These activities were not observed with the diastereomer NCB-0970, indicating the stereospecific binding of NCB-0846 to TNIK. Following its discovery as a Wnt signaling inhibitor, NCB-0846 was also shown to inhibit transforming growth factor-β (TGFβ) signaling and the epithelial-to-mesenchymal transition (EMT) of lung cancer cells [21]. The therapeutic potential of TNIK inhibition by NCB-0846 in synovial sarcoma was also demonstrated [20].

The structural basis for the stereospecific binding of NCB-0846 to TNIK was clarified by X-ray crystallography [12] (Figure 4c). The NCB-0846 quinazoline scaffold interacts with the hinge region of TNIK through two hydrogen bonds with the Cys108 backbone. The terminal hydroxy group forms an intramolecular hydrogen bond, as well as a hydrogen bond with the backbone of Gln157. These interactions appear to contribute to the stereospecific binding of NCB-0846, because NCB-0970, with the terminal hydroxyl group in the opposite configuration, showed a 13-fold decrease in TNIK inhibitory activity as compared to NCB-0846. 

A comparative structural analysis of NCB-0846 and compound **9** revealed the structural basis for the differences in their anti-Wnt signaling activities [12]. The overall structure of TNIK bound to NCB-0846 was an open, inactive conformation that overlapped well with apo-TNIK (Figure 3c), in contrast to the closed, active conformation of TNIK bound to compound **9** (Figure 3b). As shown in Figure 5a, the structural difference between the NCB-0846-bound and compound **9**-bound TNIK structures is evident in the shift of the αC-helix in the N-lobe. In the compound **9**-bound TNIK structure, the formation of the “regulatory (R)-spine”, an assembly of hydrophobic residues spanning the N- and C-lobes, was observed, which is indicative of the active conformation of kinases. However, in the NCB-0846-bound TNIK, the R-spine was disassembled due to conformational changes in Leu73 (αC-helix) and Phe172 (activation loop) (Figure 5b). These conformational changes elicited by inhibitor binding may affect the scaffold function of the full-length TNIK protein when it interacts with TCF4 and β-catenin to optimize the transcription activity.

### 3.4. Naphthyridine-Based Inhibitors

Recently, two structures of TNIK bound to naphthyridine analogs were reported (PDB ID: 6RA5, compound **9**; PDB ID: 6RA7, compound **10**) [25]. These naphthyridine analogs are nanomolar inhibitors of TNIK that were optimized from pyridopyrimidine-based MAK4K4 inhibitors [33]. They also inhibit MAP4K4 and MINK1, which share high sequence homology with TNIK [25].

The two crystal structures revealed that the naphthyridine analog interacts with the hinge region of TNIK through two hydrogen bonds with the backbone carbonyl and amide groups of Glu106 and Cys108 (Figure 4d). The naphthyridine analog also hydrogen bonds with Lys54, which forms a salt bridge with Glu69. Thus, the overall structure of TNIK bound to the naphthyridine analog was in a closed, active conformation, similar to those observed for PD407824 and phenylaminopyridine analog binding (Figure 3d). The effects of naphthyridine analogs on the cellular and biological functions of TNIK await further study.

### 3.5. ON108600

ON108600 is a multi-kinase inhibitor that simultaneously inhibits casein kinase 2 (CK2), TNIK, and DYRK1, targeting triple-negative breast cancer (TNBC) [18,19]. Although the ON108600-bound structure of TNIK has not been solved, the structure of the CK2α1–ON108600 complex was recently reported [19]. The structure revealed that the drug core, consisting of a benzothiazinone ring and an aromatic ring, closely mimics the hydrogen-bonding network of the substrate analogs, AMP-PNP and GMP-PNP, when bound to CK2α1. Furthermore, ON108600 appeared to alter the local conformation of CK2α1 and inhibit CK2 holoenzyme formation in particular, by disrupting its interaction with the regulatory CK2β subunit. The structure of the CK2α1–ON108600 complex may help predict the effect of this inhibitor on the TNIK structure.

## 4. Perspectives for the Molecular Design of Future Inhibitors

Among the above TNIK inhibitors, quinazoline-based NCB-0846 is a promising lead compound for the molecular design of future colorectal cancer therapeutics because of its anti-Wnt signaling and ant-tumorigenesis activities [12]. In addition, the dihydrobenzoxazepinone-based compound **21k** [27] and the benzoxazolone-based compound **8g** [28] have also recently been reported as promising TNIK inhibitors that inhibit colorectal cancer cell proliferation (Table 1). The chemical structures of compound **21k** and compound **8g** share the same terminal 7-azaindole moiety (Figure 2). Molecular docking showed that 7-azaindole of these inhibitors hydrogen bonds to the hinge region of TNIK and does not interact with the deepest part of the ATP-binding cleft [27,28]. Thus, it seems very likely that compound **21k** and compound **8g**, like NCB-0846, bind to the open, inactive conformation of TNIK kinase, which is essential for the anti-Wnt signaling activity of these inhibitors.

## 5. Structural Model of Full-Length TNIK Indicates Scaffold Function

The full-length structures of TNIK and its related kinases remain unsolved. However, a recent structural model of TNIK (Figure 6), obtained with the protein structure prediction tool AlphaFold [34], has provided some insight into its scaffold function. Following the kinase domain that binds to TCF4, there is an intermittent helical segment corresponding to the β-catenin binding region. Thus, it is possible that TNIK brings β-catenin into contact with TCF4, thereby enhancing the transcriptional activity of the TCF4–β-catenin complex. Given the structural dynamics of the kinase domain, which acts as a molecular switch [35], it is possible that the conformation of the active kinase domain, stabilized by ATP binding, is required for the full activation of the TCF4–β-catenin complex.

By contrast, the C-terminal CNH domain has a β-propeller structure that binds to the small GTPase Rap2 [2]. Recently, the structures of the CNH domain, which has the homologous function of binding small GTPases, were elucidated [36,37]. The CNH domain of TNIK is separated from the kinase domain and subsequent helices by an approximately 500-residue unstructured region. Thus, the CNH domain is expected to be flexible relative to the N-terminal kinase domain, which is consistent with its independent function of cytoskeletal regulation [1].

## 6. Conclusions

Recent efforts to explore TNIK inhibitors have led to the discovery of several promising compounds with potential anticancer activities. Among the inhibitors for which TNIK bound structures are available, NCB-0846 is unique in that it stabilizes the open, inactive conformation of the TNIK kinase domain. The difference between NCB-0846 (anti-Wnt signaling activity) and phenylaminopyridine analogs (no anti-Wnt signaling activity) may arise from the different conformations of the TNIK kinase domain resulting from inhibitor binding. Structural information on the binding of other inhibitors to TNIK would strengthen this hypothesis. The structural basis for the relationship between the conformational changes in the kinase domain and the formation of the active TCF4–β-catenin transcriptional complex must be clarified in the future, by structural analyses of TNIK in complex with TCF4 and β-catenin.

## Figures and Tables

**Figure 1 ijms-23-13010-f001:**
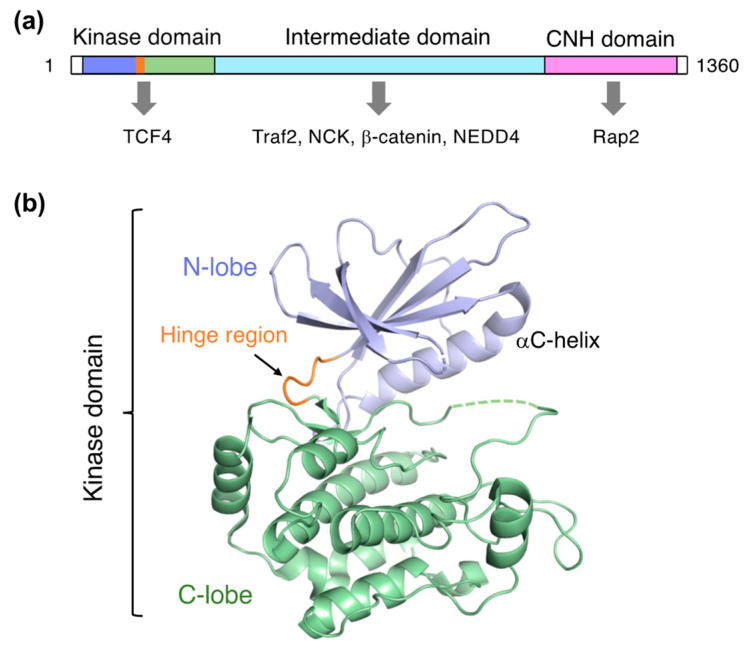
Structure of TNIK. (**a**) Domain organization of TNIK and its interacting proteins (indicated by arrows). (**b**) Crystal structure of the apo-form TNIK kinase domain (PDB ID: 5CWZ) [12].

**Figure 2 ijms-23-13010-f002:**
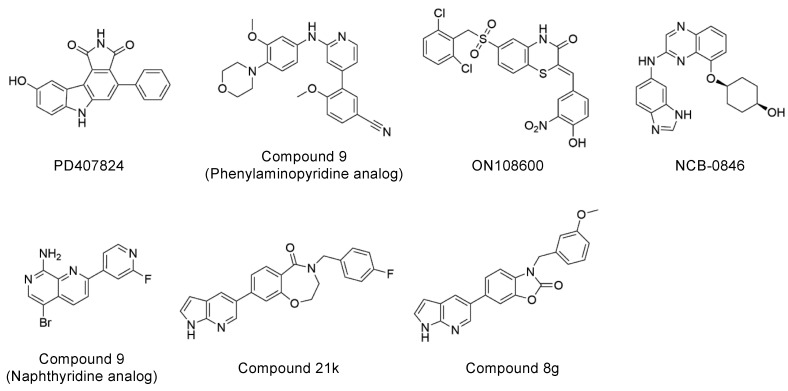
Chemical structures of representative TNIK inhibitors.

**Figure 3 ijms-23-13010-f003:**
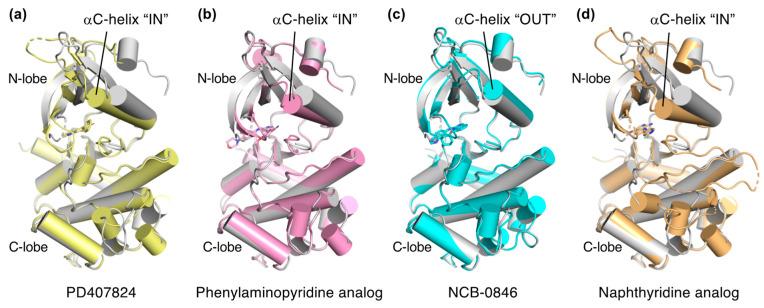
Comparison of the structures of inhibitor-bound TNIK (shown in different colors) and apo-form TNIK (gray) (PDB ID: 5CWZ) [12]. Bound inhibitors are shown as stick models. (**a**) PD407824 (PDB ID: 2X7F). (**b**) Phenylaminopyridine analog (PDB ID: 5AX9). (**c**) NCB-0846 (PDB ID: 5D7A). (**d**) Naphthyridine analog (PDB ID: 6RA5).

**Figure 4 ijms-23-13010-f004:**
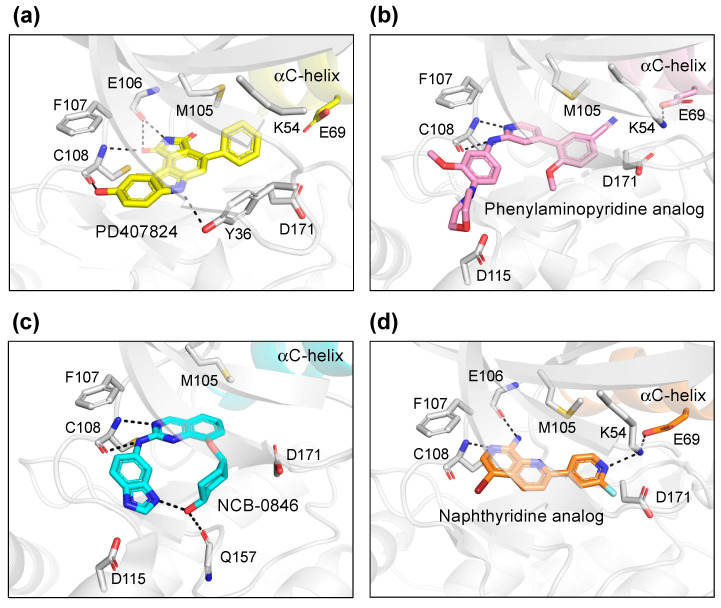
Binding modes of TNIK inhibitors. (**a**) PD407824 (PDB ID: 2X7F). (**b**) Phenylaminopyridine analog (PDB ID: 5AX9). (**c**) NCB-0846 (PDB ID: 5D7A). (**d**) Naphthyridine analog (PDB ID: 6RA5).

**Figure 5 ijms-23-13010-f005:**
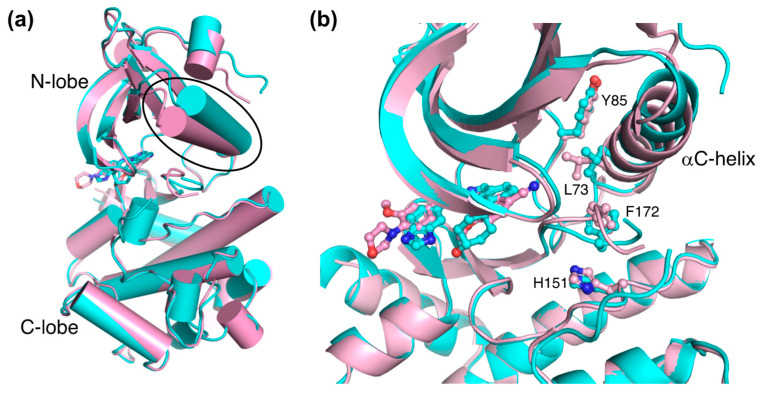
Comparison of the structures of NCB-0846-bound TNIK (cyan) (PDB ID: 5D7A) and phenylaminopyridine analog-bound TNIK (pink) (PDB ID: 5AX9). (**a**) Overall structure of the kinase domain. The different positions of the αC-helix are indicated by the oval. Bound inhibitors are shown in stick models. (**b**) Close-up view of bound inhibitors and hydrophobic R-spine residues (shown in ball-and-stick models).

**Figure 6 ijms-23-13010-f006:**
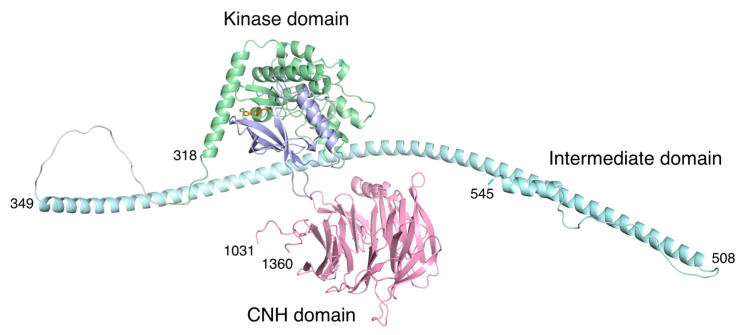
Structural model of human TNIK by AlphaFold prediction [34]. The color coding of the domains is the same as in Figure 1. The approximately 500-residue unstructured region (residues 546-1030) in the intermediate domain was omitted for clarity. Due to the unstructured regions flanking the three domains, the arrangement of each domain is expected to be flexible.

**Table 1 ijms-23-13010-t001:** TNIK inhibitors and their TNIK-bound structures.

Inhibitor	Chemical Skeleton	IC_50_ of TNIK Activity	Biological Activity	PDB ID (Conformation)
PD407824	Phenylpyrrolocarbazole	0.7 nM [9]	-	2X7F (closed/active)
NCB-0001	Aminothiazole	9 nM [14]	-	-
NCB-0005 [15]/ KY-05009	Aminothiazole	Ki = 100 nM [16]	Anti-Wnt signaling [16] Anti-TGFβ-activated EMT [16]	-
Compound **9**	Phenylaminopyridine	8 nM [17]	No anti-Wnt signaling [17] No effect on colorectal cancer cell viability [17]	5AX9 [12] (closed/active)
ON108600 [18]	Benzothiazinone	5 nM [19] (5 nM for CK2α1) (7 nM for DYRK1B)	Anti-TNBC cell growth [19] Anti-tumorigenesis in TNBC [19] Anti-TNBC metastasis [19]	-
NCB-0846	Quinazoline	21 nM [12]	Anti-Wnt signaling [12] Anti-colorectal cancer cell growth/stemness [12] Anti-tumorigenesis in colorectal cancer [12] Anti-tumorigenesis in synovial sarcoma [20] Anti-TGFβ-activated EMT [21]	5D7A [12] (open/inactive)
Mebendazole	Benzimidazole	Kd ~ 1 μM [22]	-	-
PF-794	Aminopyridine	39 nM [23]	Inhibits p120-catenin phosphorylation in cells [23]	-
Jatrorrhizine	Isoquinoline	-	Anti-Wnt signaling [24] Anti-EMT [24] Anti-mammary carcinoma cell growth [24]	-
Compounds **9**, **10**	Naphthyridine	8 nM, 7 nM [25]	Inhibits phosphorylation in cells [25]	6RA5, 6RA7 [25] (closed/active)
Compound **16**	Pyrrolopyridine	0.13 nM [26]	Inhibits IL2 secretion [26]	-
Compound **21k**	Dihydrobenzoxazepinone	26 nM [27]	Anti-Wnt signaling [27] Anti-colorectal cancer cell growth [27] Anti-tumorigenesis in colorectal cancer [27]	-
Compound **8g**	Benzoxazolone	50 nM [28]	Anti-Wnt signaling [28] Anti-colorectal cancer cell growth [28]	-

## Data Availability

Not applicable.

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
