# Peer review of "Structural Insight into TNIK Inhibition"

_ijms, 2022, doi:10.3390/ijms232113010_

Round 1

Reviewer 1 Report

After evaluating the manuscript "Structural insight into TNIK inhibition " I have to recommend its minor revision in the current version.

1. The review should be supplemented with the structures of the described inhibitors.

2. The review should be supplemented with a section considering recommendations for the molecular design of future inhibitors (focusing on prospective scaffolds for development and certain structural motives).

Author Response

Reviewer #1:

After evaluating the manuscript "Structural insight into TNIK inhibition " I have to recommend its minor revision in the current version.

1. The review should be supplemented with the structures of the described inhibitors.

As per the reviewer’s comment, we have added chemical structures of the inhibitors in a new Figure 2.  

2. The review should be supplemented with a section considering recommendations for the molecular design of future inhibitors (focusing on prospective scaffolds for development and certain structural motives).

Thank you for this suggestion. We have added a new section 4 regarding perspectives for the molecular design of future inhibitors (lines 180-192).

Reviewer 2 Report

The manuscript by Mutsuko Kukimoto-Niino and colleagues summarizes the structural insights of TNIK (TRAF2- and NCK-interacting kinase) and its inhibitors. This article provides detailed structural information on this colorectal cancer target and the inhibitor-TNIK complex. These findings may pave the way for the development of innovative colorectal cancer drugs. It is recommended that the paper be published in the International Journal of Molecular Sciences. However, I have some suggestions about the manuscript:

1. In figure 1a, the color code of those domains was better as in figure 1b and Figure 5. The color of the label "N-lobe" should be the same as the cartoon.

2. Table 1 includes some small molecular inhibitors with various chemical skeletons. It would be helpful if those inhibitors were categorized based on their structure, or chemical properties. It can be useful to understand how these inhibitors work and to design new drugs based on those findings.

3. Since artificial intelligence has been applied to the prediction of protein structure, a number of programs have been developed for protein complexes, for example, RoseTTAfold and AlphaFold Multimer. In addition, several tools are remarkable for finding binding sites using deep learning. This manuscript would be enhanced if that information were available.

Author Response

Reviewer #2:

The manuscript by Mutsuko Kukimoto-Niino and colleagues summarizes the structural insights of TNIK (TRAF2- and NCK-interacting kinase) and its inhibitors. This article provides detailed structural information on this colorectal cancer target and the inhibitor-TNIK complex. These findings may pave the way for the development of innovative colorectal cancer drugs. It is recommended that the paper be published in the International Journal of Molecular Sciences. However, I have some suggestions about the manuscript:

  1. In figure 1a, the color code of those domains was better as in figure 1b and Figure 5. The color of the label "N-lobe" should be the same as the cartoon.

We have changed the colors in Figure 1a to match those in Figure 1b and the new Figure 6.

  1. Table 1 includes some small molecular inhibitors with various chemical skeletons. It would be helpful if those inhibitors were categorized based on their structure, or chemical properties. It can be useful to understand how these inhibitors work and to design new drugs based on those findings.

As per the reviewer’s comment, we have included a column to indicate the chemical skeleton of the inhibitors in Table 1.

  1. Since artificial intelligence has been applied to the prediction of protein structure, a number of programs have been developed for protein complexes, for example, RoseTTAfold and AlphaFold Multimer. In addition, several tools are remarkable for finding binding sites using deep learning. This manuscript would be enhanced if that information were available.

Thank you for your comment. We agree that recent AI tools have the potential to identify protein binding sites in TNIK. We have performed additional calculations using AlphaFold Multimer for the TNIK-TCF4 protein complex, which is key to our discussion. However, the structure of TCF4 was largely unfolded except for the known DNA binding domain, and no reliable structural model of the TNIK-TCF4 complex was obtained. Therefore, as described in lines 223-226, future experiments, including structural analysis, will likely be needed to identify protein binding and conformational changes in TNIK.